# Carbapenem-Resistant *Klebsiella pneumoniae* Associated with COVID-19

**DOI:** 10.3390/antibiotics10050561

**Published:** 2021-05-11

**Authors:** Irina Magdalena Dumitru, Mirela Dumitrascu, Nicoleta Dorina Vlad, Roxana Carmen Cernat, Carmen Ilie-Serban, Aurelia Hangan, Raluca Elena Slujitoru, Aura Gherghina, Corina Mitroi-Maxim, Licdan Curtali, Dalia Sorina Carp, Anca Dumitrescu, Romelia Mitan, Rodica Lesanu, Sorin Rugina

**Affiliations:** 1Clinical Infectious Diseases Hospital, 100 Ferdinand Str, 900709 Constanta, Romania; mireladumitrascu33@gmail.com (M.D.); nicoleta.lalescu@yahoo.com (N.D.V.); roxana.cernat@seanet.ro (R.C.C.); carmenilieserban@yahoo.com (C.I.-S.); aura.hangan@gmail.com (A.H.); ralucaslujitoru@yahoo.com (R.E.S.); auragherghina@yahoo.com (A.G.); coramitroi@yahoo.com (C.M.-M.); licdan15@yahoo.com (L.C.); cdaliasorina@yahoo.com (D.S.C.); ancadumitrescu@ymail.com (A.D.); romeliaiuliana@yahoo.com (R.M.); rodicalesanu@yahoo.com (R.L.); sorinrugina@yahoo.com (S.R.); 2Faculty of Medicine, Ovidius University of Constanta, Aleea Universității, nr. 1, 900470 Constanta, Romania; 3Doctoral School of Medicine, Ovidius University of Constanta, Aleea Universității nr. 1, 900470 Constanta, Romania; 4Romanian Academy of Scientists, Spl. Independentei 54, 030167 București, Romania; 5Romanian Academy of Medical Sciences, Bulevardul Ion C. Brătianu 1, 030167 București, Romania

**Keywords:** COVID-19, carbapenemase, KPC, carbapenem-resistant *Klebsiella pneumoniae*

## Abstract

Infections with carbapenem-resistant *Enterobacteriaceae* are emerging as an important challenge in healthcare settings. Currently, carbapenem-resistant *Klebsiella pneumoniae* (CRKP) are the species of CRE most commonly encountered in hospitals. CRKP is resistant to almost all available antimicrobial agents, and infections with CRKP have been associated with high rates of morbidity and mortality, particularly among persons with prolonged hospitalization exposed to invasive devices. We report nine patients hospitalized in an intensive care unit (ICU) with severe coronavirus disease 2019 (COVID-19) who developed invasive infections due to carbapenemase-producing *Klebsiella pneumoniae* (CP-Kp), KPC and OXA-48, strains that have not been previously identified in our hospital. Despite ceftazidime/avibactam therapy, five patients died. Coinfections can contribute to a poor prognosis for patients with COVID-19, especially for high-risk populations such as elderly patients. Therefore, it is crucial to establish a rigorous program of antibiotic administration in intensive care units.

## 1. Introduction

Bacterial infections associated with viral infections are a major cause of morbidity and mortality. In previous influenza epidemics, secondary bacterial infections were found in 30% of cases, especially in intensive care units, and the germs most commonly associated with influenza infection were *Streptococcus pneumoniae*, *Staphylococcus aureus* and *Streptococcus pyogenes* [1]. A meta-analysis by Langford et al., published in 2020, showed that, in the case of 3338 patients hospitalized for COVID-19, a secondary bacterial infection was present in 6.9% of cases and was more common in critically ill patients (13.8%) [2]. The most common bacteria were *Mycoplasma pneumoniae* and Gram-negative germs like extended-spectrum beta-lactamase (ESBL)-positive *Klebsiella pneumoniae*, ESBL-positive *Pseudomonas aeruginosa*, carbapenem-resistant *Klebsiella pneumoniae* and highly resistant *Acinetobacter baumannii* [1].

Several Gram-negative germs such as *Klebsiella pneumoniae* spp. are asymptomatic colonizers of the human gastrointestinal tract and common opportunistic human pathogens and play an important role in resistance gene exchange and dissemination in healthcare settings [3]. *Klebsiella pneumoniae* carbapenemases KPC, encoded by bla_KPC_, and *Klebsiella pneumoniae* carbapenemases OXA-48, encoded by bla_OXA-48_, are the most common carbapenemase genes [4].

Multiple comorbidities (malignancy, congestive heart failure, chronic lung disease, chronic kidney disease and diabetes); long-term antimicrobial treatment; critical illness; various invasive devices (mechanical ventilation, urinary catheter, central vascular access, dialysis and endoscopy); transfusion events and exposure to other colonized patients are risk factors for acquisition and infection [5,6,7].

The gene that encodes the bla_KPC_ enzyme is carried on a mobile piece of genetic material (transposon), which increases the risk for dissemination. Patients with unrecognized CRKP colonization have served as reservoirs for transmission during healthcare-associated outbreaks [8].

## 2. Results

Out of 25 positive biological products for multidrug-resistant germs, CRKP were detected in nine patients only in the ICU (36%).

One out of nine (11.1%) *Klebsiella Pneumoniae* strains from COVID-19 patients were positive for the bla_OXA-48_ gene (from sputum); four out of nine (44.4%) were positive for the bla_KPC_ gene (two from sputum, one from urine and one from blood) and five COVID-19 patients (55.5%) were positive for the both genes bla_KPC_ and bla_OXA-48_ from sputum (Table 1).

The ages of the patients ranged from 47 to 75 years, the median age was 65, all the patients were hospitalized in the intensive care unit with severe (five cases) or critical (four cases) forms of COVID-19, only one patient was intubated and all other patients were ventilated noninvasively with Continuous Positive Airway Pressure (CPAP).

All patients had multiple comorbidities; the most common risk factor was hypertension (seven patients), followed by hypothyroidism (three patients), asthma and obesity (two patients each).

The administration of dexamethasone in immunosuppressive doses was common in all patients, and at least one dose of tocilizumab was used in critically ill patients, followed by anakinra for severe inflammatory syndrome.

After isolation of the bla_KPC_ gene and the bla_OXA-48_ gene, ceftazidime/avibactam was administered without a favorable outcome to those in critical condition where the diagnosis was late. In our study, five patients died; the unfavorable evolution towards death was correlated especially with ages over 70 years (median age—71.8 years), the administration of tocilizumab (four out of five patients received one dose of tocilizumab) and the presence of CRKP in the sputum and blood (bla_KPC_ expression was detected in four out of five patients).

The median duration of ICU hospitalization was 16.4 days (between 4 and 46 days), and the median of length of stay in hospital was 23.22 days (between 10 and 48 days) (Table 2).

Antibiograms show an extensive resistance to almost all the classes of antibiotics (Table 3).

## 3. Discussion

The European Centre for Disease Control (ECDC) reported that, in Europe, 7.5% of *Klebsiella pneumoniae* isolated from blood cultures were resistant to carbapenems, while, in Italy, it was 26.8% [9]. With severe COVID-19 cases, the percentage of carbapenem-resistant *Klebsiella pneumoniae* (CR-Kp) infections increased, leading to a higher mortality rate (30–70%) [10]. The spread of multidrug-resistant (MDR) bacteria, especially *Klebsiella* MDR during the COVID-19 era, was facilitated by the increased consumption of antibiotics during this period. International studies indicate that approximately 70% of hospitalized patients with COVID-19 receive antibiotics, most often with broad spectrums, despite a lack of evidence of bacterial coinfections [11].

CRE with different carbapenemase genes show variations in their geographic spread. Regions and countries considered as having the highest prevalence of the various carbapenemase-producing CRE are the United States, Israel, Greece, Italy (KPC), Turkey, the Middle East and North Africa (OXA-48) [9,12,13].

In Romania, a study published by Lixandru et al. (2015) analyzed, for the first time, the situation of CRKP infections and concluded that the carbapenemase most frequently detected is OXA-48, representing 79% of the CRKP strains [14]. Another study published in 2018 by Baicus et al. discovered that OXA-48 was the most frequently identified genotype in 73.77% of cases (45 isolates); only 1.63% (one isolate) presented bla_OXA-48_ and bla_KPC_, and only 8.19% (five isolates) presented bla_KPC_ [15]. In our study, eight out of nine patients presented KPC-Kp infections: four patients bla_KPC_ and another four patients bla_KPC_ and bla_OXA-48_.

In our hospital, before the pandemic, only OXA-48 was isolated; this was the first time that the presence of KPC CRE were reported.

Since the beginning of the pandemic, several authors have reported the association between COVID-19 and carbapenemase-secreting *Klebsiella*, stressing that these infections may have the potential for seriously complicating the course of COVID-19 [16]. In a study published by Montrucchio et al., the presence of a carbapenemase-producing coinfection with *Klebsiella pneumoniae* was correlated with the presence of comorbidities such as hypertension, asthma, smoking and obesity and the administration of corticosteroids and tocilizumab, as well as a previous administration of broad-spectrum antibiotics [17]. International studies have shown that antibiotics such as ceftriaxone, doxycycline, azithromycin, quinolones and carbapenems have been widely used for COVID-19 infections, even in the absence of obvious signs of bacterial infections [18,19].

Given that coronaviruses will be a permanent challenge for the coming years, and the administration of antibiotics has implicitly increased the rate of germ resistance, it is necessary to impose measures to actively monitor the circulation of highly resistant germs, implement appropriate strategies to limit the spread of these pathogens and a rigorous antimicrobial stewardship program in high-risk areas where COVID-positive patients are being assisted.

Alternative therapeutic approaches such as therapies with antimicrobial peptides or phages, which are in studies with promising results, should also be considered.

## 4. Materials and Methods

Nine critically ill patients admitted to the ICU wards in our hospital for SARS-CoV-2 infection were enrolled in the present study.

An etiological diagnosis was performed by a real-time polymerase chain reaction (RT-PCR) from a nasopharyngeal swab. Laboratory tests such as a complete blood count, C-reactive protein (CRP), D-dimer, clotting tests, lactic dehydrogenase (LDH), interleukin 6, ferritin and procalcitonin can identify the risk of the disease with greater severity, thromboembolic complications, myocardial damage and a worse prognosis. Thorax Computed Tomography (CT) is performed in all patients with confirmed COVID-19-induced pneumonia.

The inclusion criteria were being infected by COVID-19, hospitalized, admitted into the ICU >48 h and mechanically ventilated (CPAP). Ventilator-associated pneumonia (VAP) was identified in only one intubated person, where a urinary catheter was present. All the patients had a fever (>38 °C), leukocytosis, an elevated erythrocyte sedimentation rate, C-reactive protein (CRP) and elevated procalcitonin levels. Patients were given antibiotics such as doxycycline plus ceftriaxone or meropenem before admission to the ICU.

The *Klebsiella* strains were isolated using the AMS 2000 TRADING IMPEX culture media. The identification of the *Klebsiella* strains, testing of the antimicrobial susceptibility and characterization of the CRKP isolates was performed with the automatic systems MALDI-TOF MS 1000, VITEK 2-Compact 15 and diffusimetric, using ROSCO confirmatory discs. The antimicrobial susceptibility was interpreted in accordance with the European Committee for Antimicrobial Sensitivity Testing (EUCAST) 2021. We did not use the RT-PCR test to amplify the relevant CRE determinants KPC and OXA-48.

## 5. Conclusions

In the ICU, the prevalence of KPC-Kp infections has increased significantly in the COVID-19 period compared to the non-COVID period (3.8%). A national study is required for the antimicrobial susceptibility and molecular epidemiology of the CRKP strains.

Coinfections can contribute to a poor prognosis for patients with COVID-19, especially for high-risk populations such as elderly patients. Immunosuppressive treatments can lead to the unfavorable evolution of patients coinfected with COVID-19 and multidrug-resistant *Klebsiella*.

The results obtained show that we need to focus more on the CP-Kp infections among COVID-19 patients owing to their extreme fragility, probably linked to immunosuppressive therapy, and prolonged ICU hospitalization. Therefore, it is crucial to establish a rigorous program of antibiotic administration in intensive care units, as well as compliance with the universal precautions that lead to limiting the spread of germs.

It is clear that bacterial infections, secondary to viral infections, play a crucial role in the mortality rate of patients with viral respiratory infections, such as COVID-19; therefore, deepening these aspects through new clinical–epidemiological studies will facilitate finding effective solutions to resolve these problems.

The presence of CRKP in ICU ward effluents means that it may escape to the aquatic environment, which would be problematic for humans and the public health. This is not only a hospital burden but, also, a burden to the environment [20].

## Figures and Tables

**Table 1 antibiotics-10-00561-t001:** Patient characteristics (demographics, clinical forms of the disease, treatment and type of carbapenemases).

Pts	Sex	Age Years	Comorbidities	Clinical Forms COVID-19	ETI	Immunosuppressive Treatment	Biologic Product	CRKP	Evolution
1	F	74	heart failure, hypertension, atrial fibrillation, chronic kidney disease, anemia	critical	yes	1, 2, 3, 4	sputum	KPC, OXA-48	death
2	F	47	hypertension breast cancer, hypothyroidism	severe	no	1, 3, 4	sputum	KPC, OXA-48	good
3	F	67	myasthenia gravis, hypothyroidism	critical	no	1, 2, 3, 4	sputum	KPC, OXA-48	death
4	M	63	hypertension, hypothyroidism obesity dyslipidemia	severe	no	1, 3, 4	urine	KPC	good
5	M	75	hypertension diabetes	severe	no	1, 3, 4	sputum	KPC, OXA-48	death
6	F	70	hypertension, asthma heart failure, peripheral venous insufficiency	critical	no	1, 2, 3, 4	blood	KPC	death
7	M	61	hypertension	severe	no	1, 2, 4	sputum	KPC	good
8	F	55	hypertension, asthma, obesity	critical	no	1, 2, 3, 4	sputum	KPC	good
9	M	73	hypertension gastric ulcer chronic pancreatitis	severe	no	1, 2, 4	sputum	OXA-48	death

Legend: pts = patients, F = female, M = male and TI = endotracheal intubation. 1 = Dexamethasone, 2 = Tocilizumab, 3 = Anakinra and 4 = Hyperimmune plasma.

**Table 2 antibiotics-10-00561-t002:** Timing between hospital admission, CRPK diagnosis and treatment.

Pts	Length of Stay in Hospital (Days)	Days of ICU Intake after Hospital Admission	Length of Stay in ICU (Days)	Day after Admission in Hospital CRKP Was Diagnosed	Antimicrobial Therapy before CRKP Confirmation
1	25	7	18	9	meropenem
2	14	5	4	6	levofloxacin
3	30	the patient was hospitalized only in ICU	30	18	meropenem + linezolid
4	17	5	7	13	ceftriaxone + doxycycline
5	18	the patient was hospitalized only in ICU	18	8	meropenem
6	48	2	46	25	meropenem + colistin
7	10	3	4	11	meropenem
8	29	7	10	23	meropenem
9	18	7	11	12	ceftriaxone + doxycycline

Legend: pts = patients, CRKP—carbapenem-resistant *Klebsiella pneumoniae*. ICU—intensive care unit.

**Table 3 antibiotics-10-00561-t003:** CRPK resistance patterns.

Antibiotics	1	2	3	4	5	6	7	8	9
Ampicillin	R	R	R	R	R	R	R	R	R
Amoxicillin–Clavulanic acid	R	R	R	R	R	R	R	R	R
Piperacillin–tazobactam	R	R	R	R	R	R	R	R	R
Cefotaxime	R	R	R	R	R	R	R	R	R
Ceftazidime	R	R	R	R	R	R	R	R	R
Cefepime	R	R	R	R	R	R	R	R	R
Ertapenem	R	R	R	R	R	R	R	R	R
Imipenem	R	R	R	R	R	R	R	R	R
Meropenem	R	R	R	R	R	R	R	R	R
Amikacin	R	R	R	R	S	S	S	R	S
Gentamicin	R	R	R	R	R	S	S	R	R
Ciprofloxacin	R	R	R	R	R	R	R	R	R
Fosfomycin	R	R	R	R	R	S	R	R	R
Sulfamethoxazole–Trimethoprim	S	R	R	S	R	R	R	S	R
Ceftazidime/avibactam	S	S	S	S	S	S	S	S	S

R = resistant and S = sensitive.

## Data Availability

Restrictions apply to the availability of these data. Data are available from the authors with the permission from the Clinical Infection Diseases Hospital Constanta.

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
