# Peer review of "Carbapenem-Resistant Klebsiella pneumoniae Associated with COVID-19"

_antibiotics, 2021, doi:10.3390/antibiotics10050561_

Round 1

Reviewer 1 Report

General comments

This article describes the relationship between COVID and secondary infections with multidrug resistant bacteria. The English language is clear, but should nevertheless be further finetuned by a native speaker. The clinical context ideally should be extended by including information on the nosocomial time frame for the healthcare associated aspect (acquisition epidemiology in the hospital/ICU ; length of stay in hospital, length of stay in ICU, date of COVID onset, date of CRKP confirmation in the laboratory, antibacterial therapy type and time of administration, time of resolution/fatality).

The antimicrobial consumption as exerted in the concerned ICU should be better documented (treatment incidence expressed e.g. by DDD/patient days or admissions) to prelude the conclusion/last sentence of the abstract.

The article could be further improved by additional clinical information of the timeline and therapies applied. Phenotyping results could be included (antibiogram) to further assess relatedness of strains. Ideally, genotypic relatedness (fingerprinting by MLST, PFGE, WGS, …) of the strains(and/or transposons should shed more light on the suspicious origin of the strains within the ICU (nosocomial transmission).

Detailed comments

Line 16-17: write in the plural form – ‘infections ….are…’ (cfr line 31)

Line 23: write the Latin name in italic

Line 23, line 34-35, line 48-52, …: write the Latin names of bacteria in italic and carefully check this throughout the manuscript (including the table)

Line 38: ‘most common’ (instead of ‘commonest’)

Line 38: consider to write Gram-negative (with a hyphen, throughout the article)

Line 40: ‘Pseudomonas’ (check spelling)

Line 43: consider to rewrite ‘…are asymptomatic colonizers of the human gastrointestinal tract and common opportunistic human pathogens and play…’

Line 45 & lines 63-64: check applied subscript (KPC)

Line 55: spell out (first time abbreviation used, and then apply this abbreviation consequently): Carbapenem resistant K. pneumonia (cfr line 58-59 e& 82-83),

Table 1: include all abbreviations used in the legend of the table (pts, F, M, IOT, KPC, …)

Consider to include in the table the length of stay in hospital, length of stay in ICU, date of COVID onset, as well as date of CRKP confirmation in the laboratory, antimicrobial therapy type and moment of administration (cfr. Extension of lines 122-123 - indicate time after admission, and indicate time of admission in ICU).

Line 109-111: include the antimicrobial resistance selection pressure in the ICU, eg. By calculation for the previous time period (eg. Last five years, or at least last year or semester) the number of defined daily doses per (1000) patient days (or admissions), and concisely compare with internationally available references for ICU.

Materials & methods

Include how the COVID – SARScov2 diagnoses was done (PCR – Antigen – CT thorax scan, other…)

Ideally the antibiograms of the 9 patients should be included as annex, with inclusion of the year of EUCAST guidelines applied for the interpretation of the antimicrobial susceptibility testing.

Line 129: use the past tense throughout the entire section.

Author Response

This article describes the relationship between COVID and secondary infections with multidrug resistant bacteria. The English language is clear, but should nevertheless be further finetuned by a native speaker. The clinical context ideally should be extended by including information on the nosocomial time frame for the healthcare associated aspect (acquisition epidemiology in the hospital/ICU ; length of stay in hospital, length of stay in ICU, date of COVID onset, date of CRKP confirmation in the laboratory, antibacterial therapy type and time of administration, time of resolution/fatality).

The antimicrobial consumption as exerted in the concerned ICU should be better documented (treatment incidence expressed e.g. by DDD/patient days or admissions) to prelude the conclusion/last sentence of the abstract.

The article could be further improved by additional clinical information of the timeline and therapies applied. Phenotyping results could be included (antibiogram) to further assess relatedness of strains. Ideally, genotypic relatedness (fingerprinting by MLST, PFGE, WGS, …) of the strains(and/or transposons should shed more light on the suspicious origin of the strains within the ICU (nosocomial transmission).

Line 16-17: write in the plural form – ‘infections ….are…’ (cfr line 31)

            Line 16-17, infections, are

Line 23: write the Latin name in italic

– line 23 - italic

Line 23, line 34-35, line 48-52, …: write the Latin names of bacteria in italic and carefully check this throughout the manuscript (including the table)

            I write the Latin names of bacteria in italic, 23, 43, 45, 45, 46, 47, , 60, table

Line 38: ‘most common’ (instead of ‘commonest’) – line 38

Line 38: consider to write Gram-negative (with a hyphen, throughout the article)

  • Yes, line 38

Line 40: ‘Pseudomonas’ (check spelling) – yes, line 40

Line 43: consider to rewrite ‘…are asymptomatic colonizers of the human gastrointestinal tract and common opportunistic human pathogens and play…

            I rewrite – line 43-44

Line 45 & lines 63-64: check applied subscript (KPC)

            Yes, applied KPC , line 45, 63,64

Line 55: spell out (first time abbreviation used, and then apply this abbreviation consequently): Carbapenem resistant K. pneumonia (cfr line 58-59 e& 82-83),

            Abreviation CRKP- line 58, table 1 (first line), line 91, table 2,

Table 1: include all abbreviations used in the legend of the table (pts, F, M, IOT, KPC, …

Yes, legend 66-68)

Consider to include in the table the length of stay in hospital, length of stay in ICU, date of COVID onset, as well as date of CRKP confirmation in the laboratory, antimicrobial therapy type and moment of administration (cfr. Extension of lines 122-123 - indicate time after admission, and indicate time of admission in ICU).

            Table 2, 92-93

Line 109-111: include the antimicrobial resistance selection pressure in the ICU, eg. By calculation for the previous time period (eg. Last five years, or at least last year or semester) the number of defined daily doses per (1000) patient days (or admissions), and concisely compare with internationally available references for ICU.T

Table 3, LINE 119,

Materials & methods

Include how the COVID – SARScov2 diagnoses was done (PCR – Antigen – CT thorax scan, other…)

  • 142-147, TABLE 3

Ideally the antibiograms of the 9 patients should be included as annex, with inclusion of the year of EUCAST guidelines applied for the interpretation of the antimicrobial susceptibility testing.

Line 129: use the past tense throughout the entire section. - YES

Reviewer 2 Report

Irina Magdalena Dumitru and Co-workers' manuscript describes CRE infection's importance in hospitalized patients associated with COVID 19. In my opinion, such studies must be carried to implement appropriate measures in the hospital environment. The authors did a good job. However, there are some minor corrections, and with that, this manuscript may be published in this journal. 

Please below find my comments. 

Line no 39: Klebsiella pneumoniae to “Klebsiella pneumoniae”

Line no 40: Klebsiella pneumoniae to Klebsiella pneumoniae”

And same should be followed in the whole manuscript.

CRE presence in ICU wards effluents may escape to the aquatic environment and that again problematic to human and public health. This is not only a hospital burden but also to the environment. Hence, authors may cite an additional appropriate reference "https://pubmed.ncbi.nlm.nih.gov/31581701/" in the introduction section.

Did authors use PCR test to amplify the concern CRE determinants "KPC, OXA-48"? If so, please specify in the manuscript with primer pairs used. 

Author Response

Irina Magdalena Dumitru and Co-workers' manuscript describes CRE infection's importance in

hospitalized patients associated with COVID 19. In my opinion, such studies must be carried to

implement appropriate measures in the hospital environment. The authors did a good job. However,

there are some minor corrections, and with that, this manuscript may be published in this journal.

Line no 39: Klebsiella pneumoniae to “Klebsiella pneumoniae” -yes, line 39 italic

 Line no 40: Klebsiella pneumoniae to Klebsiella pneumoniae” – yes , line 40 italic

 And same should be followed in the whole manuscript., italic, line42, 45, 60, 99, 114, 125,

 CRE presence in ICU wards effluents may escape to the aquatic environment and that again problematic to human and public health. This is not only a hospital burden but also to the environment. Hence, authors may cite an additional appropriate reference "https://pubmed.ncbi.nlm.nih.gov/31581701/" in the introduction section.

  • Lie 179-181, reference 20, line247

Did authors use PCR test to amplify the concern CRE determinants "KPC, OXA-48"? If so, please specify in the manuscript with primer pairs used.

            Line 159-160 we did not use the RT-PCR test to amplify the concern CRE determinants "KPC, OXA-48

Reviewer 3 Report

Dear Authors,

the paper is well written and the highlights the importance of antimicrobial stewardship programs to control CRKP infections' spread during the COVID-19 pandemic.

I have not major comments.

Minor comments:

  • In the Results the number of patients' deaths should be reported;
  • You refer about a coloration between death and age, tocilizumab and CRKP isolation.  What kind of relationship is it? Has a statistical test been carried out?
  • Please consider the following references:
    1) Salsano A, Giacobbe DR, Sportelli E, Olivieri GM, Brega C, Di Biase C, Coppo E, Marchese A, Del Bono V, Viscoli C, Santini F. Risk factors for infections due to carbapenem-resistant Klebsiella pneumoniae after open heart surgery. Interact Cardiovasc Thorac Surg. 2016 Nov;23(5):762-768. doi: 10.1093/icvts/ivw228. Epub 2016 Jul 1. PMID: 27371609.
    2) Giacobbe DR, Salsano A, Del Puente F, Campanini F, Mariscalco G, Marchese A, Viscoli C, Santini F. Reduced Incidence of Carbapenem-Resistant Klebsiella pneumoniae Infections in Cardiac Surgery Patients after Implementation of an Antimicrobial Stewardship Project. Antibiotics (Basel). 2019 Aug 28;8(3):132. doi: 10.3390/antibiotics8030132. PMID: 31466372; PMCID: PMC6783823.

Author Response

  1. the paper is well written and the highlights the importance of antimicrobial stewardship programs to control CRKP infections' spread during the COVID-19 pandemic.

I have not major comments.

Minor comments:

  • In the Results the number of patients' deaths should be reported;
    • Line 82 (in our study five pts died)
  • You refer about a coloration between death and age, tocilizumab and CRKP isolation.  What kind of relationship is it? Has a statistical test been carried out?
    • NO STATISTICAL TEST BECAUSE only 9 patients, but in table 1, age of patients who died was 74, 67, 75, 70, 73, all 5 patients received tocilizumabi
  • Please consider the following references:
    1) Salsano A, Giacobbe DR, Sportelli E, Olivieri GM, Brega C, Di Biase C, Coppo E, Marchese A, Del Bono V, Viscoli C, Santini F. Risk factors for infections due to carbapenem-resistant Klebsiella pneumoniae after open heart surgery. Interact Cardiovasc Thorac Surg. 2016 Nov;23(5):762-768. doi: 10.1093/icvts/ivw228. Epub 2016 Jul 1. PMID: 27371609.
    2) Giacobbe DR, Salsano A, Del Puente F, Campanini F, Mariscalco G, Marchese A, Viscoli C, Santini F. Reduced Incidence of Carbapenem-Resistant Klebsiella pneumoniae Infections in Cardiac Surgery Patients after Implementation of an Antimicrobial Stewardship Project. Antibiotics (Basel). 2019 Aug 28;8(3):132. doi: 10.3390/antibiotics8030132. PMID: 31466372; PMCID: PMC6783823.
    • Yes, i considered – references 12, 13 line 226, 229

Round 2

Reviewer 1 Report

  • include CRKP in full, also in the main document (not only in the abstract).
  • delete the exact date of admission in Table 2, but transform into: length of stay, days of ICU intake after hospital admission, length of ICU stay, day after admission in hospital CRKP was diagnosed.
  • make the titles of the tables more clear for a reader (eg include objective, time frame and what information can be found in the title. 
  • check in the legend all abbreviations used (eg CRKP in table  2)
  • delete pts in table 3

Author Response

include CRKP in full, also in the main document (not only in the abstract). – line 114, 115, 157, 165, 180

delete the exact date of admission in Table 2, but transform into: length of stay, days of ICU intake after hospital admission, length of ICU stay, day after admission in hospital CRKP was diagnosed

  • Table 2, line 90

make the titles of the tables more clear for a reader (eg include objective, time frame and what information can be found in the title.

  • Table 2, line 90, 

    Table 2.  Timing between hospital admission, CRPK diagnosis and treatment

    table 3, line 96 CRPK resistance patterns

Legend Table 2,  Line 94,95

 I   deleted pts in table 3